# Prediction of Blast-Induced Ground Vibration at a Limestone Quarry: An Artificial Intelligence Approach

Clement Kweku Arthur [1], Ramesh Murlidhar Bhatawdekar [2,*], Edy Tonnizam Mohamad [2], Mohanad Muayad Sabri Sabri [3], Manish Bohra [4], Manoj Khandelwal [5] and Sangki Kwon [6,*]

1 Department of Mining Engineering, Faculty of Mining and Minerals Technology, University of Mines and Technology, Tarkwa P.O. Box 237, Ghana
2 Centre of Tropical Geoengineering (GEOTROPIK), School of Civil Engineering, Faculty of Engineering, University Teknologi Malaysia, Johor Bahru 81310, Malaysia
3 Centre of Peter the Great St. Petersburg Polytechnic University, 195251 St. Petersburg, Russia
4 Shree Cement, Beawar 305 901, India
5 Institute of Innovation, Science and Sustainability, Federation University Australia, Ballarat, VIC 3350, Australia
6 Department of Energy Resources Engineering, Inha University, Incheon 22212, Korea
* Correspondence: rmbhatawdekar2@graduate.utm.my (R.M.B.); kwonsk@inha.ac.kr (S.K.)

**Abstract:** Ground vibration is one of the most unfavourable environmental effects of blasting activities, which can cause serious damage to neighboring homes and structures. As a result, effective forecasting of their severity is critical to controlling and reducing their recurrence. There are several conventional vibration predictor equations available proposed by different researchers but most of them are based on only two parameters, i.e., explosive charge used per delay and distance between blast face to the monitoring point. It is a well-known fact that blasting results are influenced by a number of blast design parameters, such as burden, spacing, powder factor, etc. but these are not being considered in any of the available conventional predictors and due to that they show a high error in predicting blast vibrations. Nowadays, artificial intelligence has been widely used in blast engineering. Thus, three artificial intelligence approaches, namely Gaussian process regression (GPR), extreme learning machine (ELM) and backpropagation neural network (BPNN) were used in this study to estimate ground vibration caused by blasting in Shree Cement Ras Limestone Mine in India. To achieve that aim, 101 blasting datasets with powder factor, average depth, distance, spacing, burden, charge weight, and stemming length as input parameters were collected from the mine site. For comparison purposes, a simple multivariate regression analysis (MVRA) model as well as, a nonparametric regression-based technique known as multivariate adaptive regression splines (MARS) was also constructed using the same datasets. This study serves as a foundational study for the comparison of GPR, BPNN, ELM, MARS and MVRA to ascertain their respective predictive performances. Eighty-one (81) datasets representing 80% of the total blasting datasets were used to construct and train the various predictive models while 20 data samples (20%) were utilized for evaluating the predictive capabilities of the developed predictive models. Using the testing datasets, major indicators of performance, namely mean squared error (MSE), variance accounted for (VAF), correlation coefficient ($R$) and coefficient of determination ($R^2$) were compared as statistical evaluators of model performance. This study revealed that the GPR model exhibited superior predictive capability in comparison to the MARS, BPNN, ELM and MVRA. The GPR model showed the highest VAF, $R$ and $R^2$ values of 99.1728%, 0.9985 and 0.9971 respectively and the lowest MSE of 0.0903. As a result, the blast engineer can employ GPR as an effective and appropriate method for forecasting blast-induced ground vibration.

**Keywords:** artificial intelligence; backpropagation neural network; blast-induced ground vibration; Gaussian process regression





## 1. Introduction

Ground vibration is one of the main adverse blasting outcomes that has received significant attention in the mining and civil industries [1,2]. Ground vibration is known to have a lot of adverse impacts on the environment (cracks on building structures) and the stability of pit walls. It is worth mentioning that several factors contribute to the occurrence of these blast-induced ground vibrations. These factors can be categorized into controllable factors and uncontrollable factors [3,4]. The controllable factors are those that the blast engineer has control over and can change. These include the blast design parameters of stemming length, hole depth, spacing, burden, hole inclination and explosive parameters of delay timings, a maximum charge per delay, and total charge. The uncontrollable factors are those the blast engineer has no control over, and they include both geotechnical and geomechanical parameters such as rock strength, faults, and folds [5–9]. The peak particle velocity (PPV) is the index for assessing ground vibration induced by blasting [10]. When detonation of explosives takes place, high energy is released in the blast hole which fractures the rock surrounding the blasthole [11]. Some of the energy released is used to fragment and displace the rock mass. The rest of the energymove through the ground as ground vibration andimpacts surrounding structures.

Due to the adverse impact of blast-induced ground vibration, it has always been in the interest of the blast engineer to model and predicts its occurrence to minimize vibration level as much as possible. In that regard, a lot of research has been conducted since the 1950s [12] to develop models for predicting ground vibration arising out of blasting operations. These models have been developed using empirical techniques through to the use of artificial intelligence (AI) techniques [13]. These AI techniques have been found to produce more accurate results than the empirical techniques and hence have received worldwide attention due to their unique capabilities [14]. AI techniques that have been developed and used in the prediction of blasting outcomes (ground vibration, air overpressure, and flyrock) are outlined in Table 1. It is worth noting that all abbreviations used in this work are presented in the Abbreviations Section.

**Table 1.** AI Models developed and applied to predict ground vibration, air overpressure and flyrock.

| References | Methods | Application |
|---|---|---|
| [3,15–28] | FL, SVR, ANFIS, ANN, CART, GPR, ICA, SVM, ELM, GEP, PSO, BN | Ground Vibration Prediction |
| [29–33] | PSO-ANN, FIS, ANN, ICA_ANN, BIENN, GP, M5DT, SVM, KNN, CHAID | Air Overpressure Prediction |
| [34–39] | PSO-ANN, RF, BN, BBO-ELM, ORELM, ELM, WOA-SVM, GP | Flyrock |

More recently in ground vibration studies, other researchers have applied evolutionary and metaheuristic optimization algorithms to optimize simple AI techniques. Some of these works are presented in Table 2.

**Table 2.** Hybrid Models developed and applied to predict ground vibrations.

| References | Hybrid Models |
|---|---|
| [40–54] | PSO-ANN, ICA-ANN, ABC-ANN, PSO-ANFIS, ICA-FIS, FFA-ANN, GA-ANFIS, PSO-ANFSI, PSO-XGBoost, GA-SVR, PSO-SVR, FFA-SVR, GA-ANN, GWO-RVR, BAT-RVR, HHOA-RF, ICA-XGBoost, ICA-M5DT, HHOA-ELM, GOA-ELM |

Table 3 provides a detailed summary of some research on ground vibration prediction.

**Table 3.** Input parameters, size of data and AI techniques for prediction of ground vibration.

| References | Technique | Input Parameters | | | | No. of Datasets | $R^2$ |
| --- | --- | --- | --- | --- | --- | --- | --- |
| | | **Rock Mass** | **Blast Design** | **Explosives** | **Other** | | |
| [55] | ANN | ν, BI, E, Pv | HD, B, S | VOD, Q | H | 154 | 0.9864 |
| [15] | FIS | - | - | Q | H | 33 | 0.92 |
| [56] | ANN | | HD | Q | H | 162 | 0.9493 |
| [57] | SVM, ANN | - | - | Q | H | 37 | SVM = 0.89, ANN = 0.85 |
| [16] | FIS | - | B, S, ST | Q | H | 120 | 0.95 |
| [58] | ANN | - | - | Q | H | 20 | 0.93 |
| [40] | ANN-PSO | RD | B, S, N, HD, SD | Q | H | 44 | 0.94 |
| [59] | ANN | | ST, HD | Q | H | 69 | 0.957 |
| [60] | ANN | - | HD, ST | Q | H | 115 | 0.98 |
| [28] | ANN-PSO | RQD | ST, BS, SD | PF, Q | H | 88 | 0.89 |
| [61] | GA-ANN, ANFIS | - | - | Q | H, RD | 70 | GA-ANN = 0.988, ANFIS = 0.92 |
| [62] | WNN, GMDH, ANN | - | HD, NH | PF, Q | H | 210 | WNN = 0.712, GMDH = 0.684, ANN = 0.729 |
| [63] | GP, RSM, MARS | | | Q | H | 200 | GP = 0.7864, RSM = 0.7832, MARS = 0.8056 |
| [64] | ANFIS | - | - | Q | H | 90 | 0.983 |
| [65] | ANN | - | - | Q | H | 68 | 0.955 |
| [66] | ANN | | | PF, Q | H | 88 | 1 |
| [22] | GPR, ANN | - | HD, NH | PF, Q | H | 210 | GPR = 0.695, ANN = 0.688 |
| [49] | SaDE-ELM, ELM, ANN | - | HD, NH | PF, Q | H | 210 | SaDE-ELM = 0.759, ELM = 0.728, ANN = 0.729 |
| [67] | MARS, ANN | - | HD, NH | PF, Q | H | 210 | MARS = 0.7074, ANN = 0.6879 |
| [68] | LSSVM, ANN | - | HD, NH | PF, Q | H | 210 | LSSVM = 0.73, ANN = 0.729 |

Nevertheless, the application of single AI techniques is still of interest in this ever-growing technological world. ANN has been developed by [69] to predict the earth surface deformation. Thus, the predictive capacities of three artificial intelligence algorithms, backpropagation neural network (BPNN), ELM, and GPR, are investigated in this study using blasting data from a quarry (Ras Limestone Mine of Shree Cement) in India to estimate PPV values. A multivariate adaptive regression spline (MARS) approach, as well as a multivariate regression analysis (MVRA) model, was developed and used for comparison purposes. Studies have been made to compare the GPR and BPNN [22], MARS and BPNN [67], ELM and BPNN [70], GP and MARS [63], GPR and MVRA [71] and BPNN and MVRA [55]. However, little has been done in the literature to compare the predictive performance of GPR, MARS, BPNN, ELM and MVRA in ground vibration prediction studies. In that regard, this study is exploratory. It is worth mentioning that the empirical models developed for predicting blast-induced ground vibration were not considered

in this study. The reason is that studies done by [17,40,53,57,59,72,73] have proved that these empirical models do not produce accurate results. The models used in this study consider seven effective parameters, namely the average depth, a maximum charge per delay, powder factor, spacing, burden, distance and stemming length, because, as shown in [5–7], they significantly affect the intensity of ground vibration.

## 2. Study Site and Data Description

The Ras Limestone Mine of Shree Cement is located 30 km from Beawar City, Ajmer District, Rajasthan, India. The mining concession of 750.0 ha lies between longitude E 74°10′5.96″ to E 74°11′9.62″ and latitudes N 26°16′57.13″ to N 26°15′36.23″, on toposheet No. 45 J/3 & 45 J/4 of the survey of India.

The projected production capacity of the mine is 25.3 million tons of limestone per year. The mining area is generally rocky with no overburden. A general strike of limestone at Ras Mine is North-South direction and dips in the eastern direction. Limestone has four major folds and one reverse fault. Limestone strata are massive, blocky and fractured in different portions of the deposit. HRB 150 (INDUS Make) drills are used for drilling hole diameter of 165 mm. ANFO with cast booster/slurry explosives and nonel detonators are used as explosives for blasting limestone. Figure 1 shows a blasting round view with Figure 2 showing the close-up view of blasted limestone at Shree Cement Ras Limestone Mine in India.

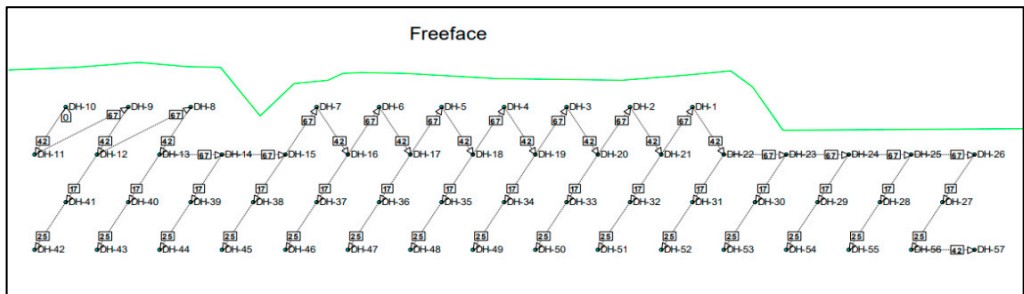

**Figure 1.** Blasting Round View.

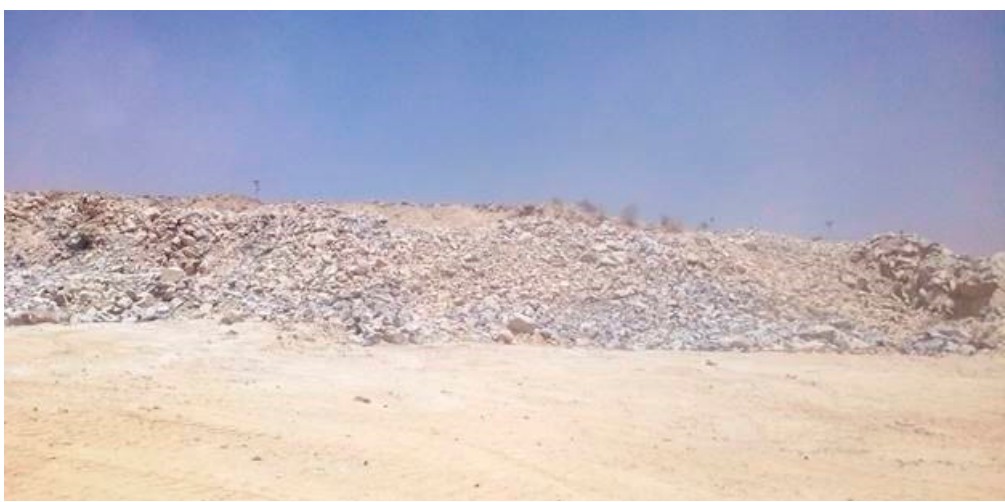

**Figure 2.** Close-up View of Blasted Limestone at Shree Cement Ras Limestone Mine in India.

As a part of this study, for the establishment of the various models described therein, a total of 101 sets of data were collected from the Ras limestone mine. The data collected consisted of parameters such as average depth (m), spacing (m), burden (m), powder factor (t/kg), the distance between the blasting point and the monitoring station (m), stemming length (m), a maximum charge per delay (kg) and PPV (mm/s). In the creation of the various models, the input parameters were average depth (m), spacing (m), burden (m),

powder factor (t/kg), the distance between the blasting site and the monitoring station (m), stemming length (m), and maximum charge per delay (kg), while the output parameter was PPV. Table 4 shows the statistical description of the dataset collected.

**Table 4.** Description of dataset parameters.

| Parameter | Category | Symbol | Units | Minimum | Average | Maximum | Standard Deviation |
|---|---|---|---|---|---|---|---|
| Average depth | | AD | m | 7.76 | 11.88 | 14.46 | 1.64 |
| Burden | | B | m | 4.5 | 4.54 | 5.5 | 0.15 |
| Spacing | | S | m | 5.5 | 6.02 | 7 | 0.31 |
| Distance | Inputs | D | m | 250 | 1356.44 | 4150 | 906.21 |
| Powder factor | | PF | t/kg | 5.38 | 6.30 | 7.83 | 0.44 |
| Stemming length | | SL | m | 3 | 3.48 | 4 | 0.29 |
| Maximum charge per Delay | | MC | kg | 73 | 129.49 | 180 | 22.94 |
| Peak Particle Velocity | Output | PPV | mm/s | 0.7 | 4.08 | 15.19 | 3.16 |

The values for the maximum charge per delay, stemming length, powder factor, spacing, burden, and average depth as statistically described in Table 2 were obtained from the daily blast plans of the mine. The distance values were calculated using the coordinates of the blasting face and monitoring locations obtained using a Global Positioning System (GPS). As shown in Figure 3, the PPV values were monitored using an Instantel Micromate ISEE Std/XM seismograph [74].

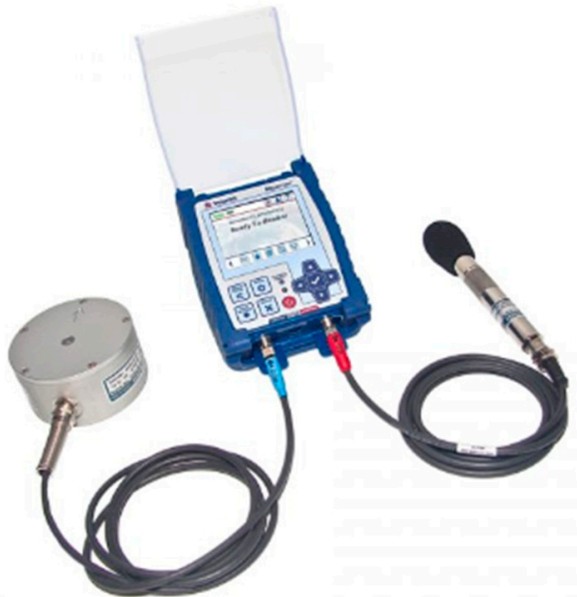

**Figure 3.** Instantel Micromate ISEE Std/XM seismograph.

It is worth mentioning that the mine has no permanent monitoring location due to different blasting positions. Thus, in monitoring the ground vibration due to blasting, the seismograph is positioned using pegs with an arrow on the geophone pointing towards the blast site. Figure 4 shows the portable monitoring station used by the mine. It is worth noting that the terrain of the Ras Limestone Mine is generally hilly.

The correlation coefficient matrix shows how strong the interaction between the input parameters (average depth, burden, spacing, distance, powder factor, stemming length, and maximum charge per delay) and the measured PPV is, as shown in Table 5.

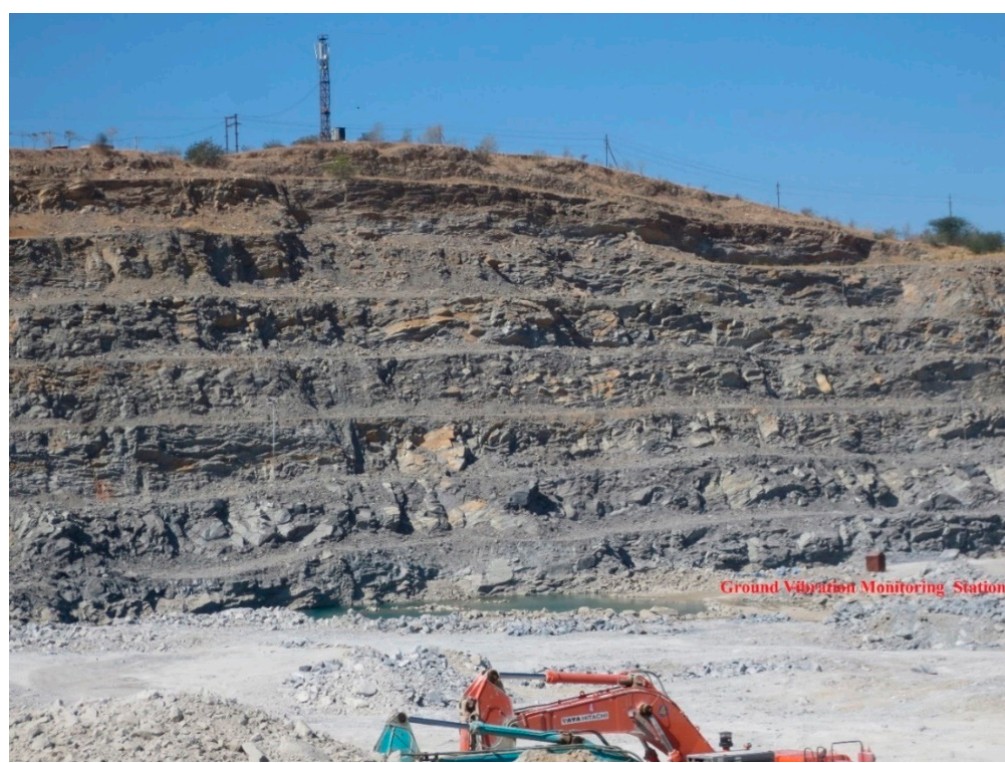

**Figure 4.** Portable ground vibration monitoring station in realistic conditions (at Shree Cement Ras Limestone Mine in India).

**Table 5.** Matrix of Correlation Coefficients Between Input Parameters and PPV Measured.

|      | AD      | B       | S       | D       | PF      | SL      | MC     | PPV |
|------|---------|---------|---------|---------|---------|---------|--------|-----|
| AD   | 1       |         |         |         |         |         |        |     |
| B    | 0.1697  | 1       |         |         |         |         |        |     |
| S    | 0.3347  | 0.8330  | 1       |         |         |         |        |     |
| D    | 0.1407  | −0.0467 | −0.0201 | 1       |         |         |        |     |
| PF   | −0.1046 | 0.5294  | 0.4905  | −0.1069 | 1       |         |        |     |
| SL   | 0.7702  | −0.1507 | −0.0235 | 0.0595  | −0.1329 | 1       |        |     |
| MC   | 0.9301  | 0.3514  | 0.4996  | 0.1617  | −0.2019 | 0.6145  | 1      |     |
| PPV  | −0.0016 | 0.1492  | 0.2160  | −0.7503 | 0.0789  | −0.0837 | 0.0293 | 1   |

## 3. Methodology

In this section, the mathematical description of the different methods applied in this study will be briefly outlined. Furthermore, the procedure followed to develop the various models as well as the models' performance indicators will be outlined.

### 3.1. Study Steps

A systematic methodology was utilized in this study. First, the data collected were prepared by removing all outliers and then were partitioned into two sets (training set and testing set) and normalized into the interval [–1,1]. Then the various models were built by selecting the model's hyperparameter. The models were then trained using the training dataset. Finally, the model's results were assessed based on the test dataset by some performance indicators. The performance results were then analyzed to either finetune the model's hyperparameter or select the model as optimum. Figure 5 shows the flowchart applied in this study.

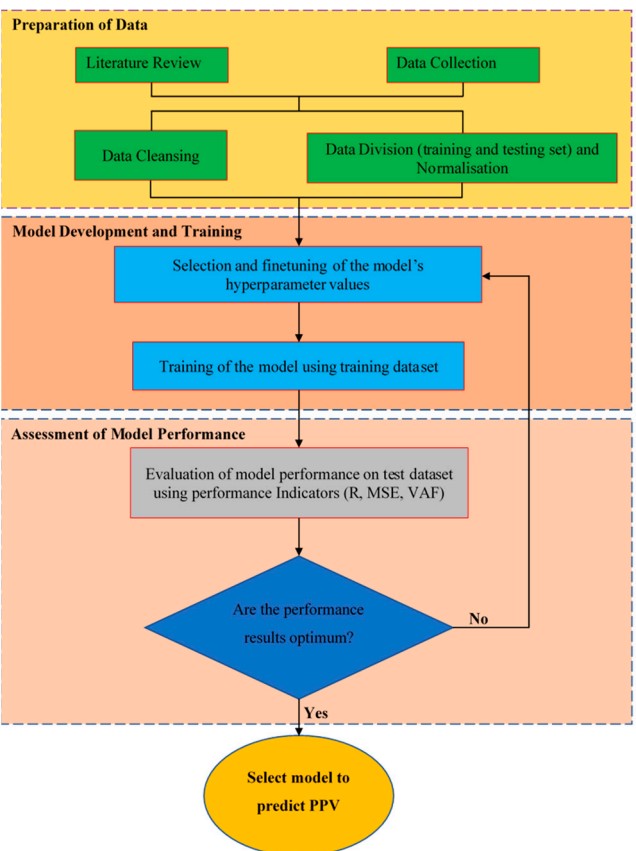

**Figure 5.** A Systematic Flowchart for Prediction of Blast-Induced Ground Vibration.

### 3.2. Mathematical Description of the Different Methods

3.2.1. GPR

Gaussian Process (GP)

GP is a nonparametric Bayesian technique that is used in regression modelling [75]. This GP process can be described as a finite assemblage of a set of arbitrary parameters that follow a multivariate Gaussian (normal) distribution [76]. That is, for every given input point from a set of input vectors $r = (r_1, r_2, r_3, \ldots, r_m)$, the probability distribution over its function $h(r)$ follows a Gaussian distribution. Thus, a GP $h(r)$ is precisely shown in Equation (1) as:

$$h(r) \sim \mathrm{GP}\big(b(r), g\big(r, r'\big)\big) \tag{1}$$

From Equation (1) it can be deduced that a GP is fully characterized by a covariance function $g(r, r')$ and a mean function (MF) $b(r)$ as expressed in Equation (2).

$$\begin{cases} b(r) = E[h(r)] \\ g(r, r') = E[(b(r) - h(r))(b(r') - h(r'))] \end{cases} \tag{2}$$

For the basic GPR, the MF is normally set as 0, however, there many other MFs which can be applied in building the GPR model [77]. The noted MFs in literature have been categorized into two kinds, namely: simple and composite. The simple MFs include zero, one, constant, linear, polynomial, nearest neighbor MFs etc. whereas the composite ones include: the scaled version, sum, product, power and warped MFs [77]. It is worth noting that this study adopted an MF with a constant, $b$.

The covariance function on the other hand is the main component in the development of the GPR model. The best covariance function is dependent on the data being modelled. Literature is replete with a number of these covariance functions [70]. However, the notable ones include: the rational quadratic, matérn class, squared exponential and the exponential

covariance functions. The most often used covariance function is the squared exponential covariance function [77,78].

Prediction Using GP

In the case of a regression modelling problem, an output variable $q$ can be approximated, given function $h(r)$ with an additive noise $\varepsilon_i$ component inherent in the dataset as shown in Equation (3).

$$q_i = h(r_i) + \varepsilon_i \tag{3}$$

Assuming this noise component $\varepsilon_i$ has a zero mean and variance $\sigma_n^2$, the prior on the noisy data is expressed in Equation (4) as:

$$\text{cov}(q) = g(r,r') + \sigma_n^2 I_n \tag{4}$$

where $I_n$ is a matrix of the n-dimensional unit.

The GP $h(r)$ (see Equation (1)) is then precisely considered in Equation (5) as:

$$h(r) \sim \text{GP}\left(b(r), g(r,r') + \sigma_n^2 I\right) \tag{5}$$

It should be emphasized that the GP model training, seeks to ascertain the best possible hyperparameter set $\Theta = \left[\beta, \chi, v_s^2, \sigma_n^2\right]$ that best fits the data sets. This can be done by the use of a maximum possible method [69] in which the log-likelihood function is maximized (Equation (6)).

$$\log(p(q|r,\Theta)) = \frac{1}{2}\log\left(\det\left(g(r,r') + \sigma_n^2 I\right)\right) - \frac{1}{2}q^T\left(g(r,r') + \sigma_n^2 I\right)^{-1}q - \frac{n}{2}\log 2\pi \tag{6}$$

Of all the maximum likelihood functions available, the conjugate gradient method is the most widely used [79] and hence was used in this study. It finds the optimal hyperparameter sets by using the partial differential of the log-likelihood function (Equation (6)) in relation to the hyperparameter set, $\Theta$ as shown in Equation (7).

$$\begin{aligned}\frac{\partial}{\partial\Theta_i}\log(p(q|r,\Theta)) &= \frac{1}{2}q^T G^{-1}\frac{\partial G}{\partial\Theta_i}G^{-1}q - \frac{1}{2}tr\left(G^{-1}\frac{\partial G}{\partial\Theta_i}\right)\\ &= \frac{1}{2}tr\left(\left(\beta\beta^T - G^{-1}\right)\frac{\partial G}{\partial\Theta_i}\right)\end{aligned} \tag{7}$$

where $\beta = G^{-1}q$ and $G = g(r,r')$.

Given the joint prior distribution of the training output variable, $q$ at point a and the value $q_*$ to be predicted at the test point $r_*$ expressed in Equation (8), the GPR model is able to predict $q_*$ by calculating the posterior distribution $p(q_*|r,q,r_*)$ (Equation (9)).

$$\begin{bmatrix} q \\ q_* \end{bmatrix} \sim GP\left(\begin{bmatrix} b(r) \\ b(r_*) \end{bmatrix}, \begin{bmatrix} g(r,r) + \sigma_n^2 I & g(r,r_*) \\ g(r_*,r) & g(r_*,r_*) \end{bmatrix}\right), \tag{8}$$

$$p(q_*|r,q,r_*) \sim GP(\bar{q}_*, cov(q_*)), \tag{9}$$

Here $\bar{q}_*$ (Equation (10)) is the mean value which is the estimation of $q_*$ and $cov(q_*)$ (Equation (11)) is the predictive variance matrix of the test data, which reveals the credibility of the prediction values [79].

$$\bar{q}_* = b(r_*) + g(r_*,r)\left[g(r,r) + \sigma_n^2 I\right]^{-1}(q - b(r)) \tag{10}$$

$$cov(q_*) = g(r_*,r_*)\left[g(r,r) + \sigma_n^2 I\right]^{-1}g(r,r_*) \tag{11}$$

### 3.2.2. BPNN

BPNN is a widely used AI technique that was developed to mimic the human brain. In this, there is an input layer that takes impulses from the outside environment as inputs to the network. These inputs $x_k$ are weighted by connecting weights $w_k$ and relayed to the hidden layer. The hidden layer contains processing units called neurons which transform the weighted input by a transfer function, $t$. It is noteworthy that biases $b$ are added to the transfer function before the transformation process. The hidden layer's output is subsequently conveyed to the output layer, which is transformed by a transfer function operating inside the hidden layer. The network's predicted values are then derived from the output, $\hat{y}$ from the output layer as shown in Equation (12).

$$\hat{y} = t\left(\sum_{k=1}^{m} w_k x_k + b\right) \tag{12}$$

In training the BPNN, a training algorithm is used in updating weights and biases based on the backpropagation error, $e$ (divergence in true and predicted value) as shown in Equation (13) so as to produce a network with a minimum propagation error.

$$e = y - \hat{y} \tag{13}$$

Several training algorithms have been developed for such purposes. However, the Levenberg–Marquardt algorithm [80] is the widely used training function due to its high convergence speed and accuracy and thus was used in this study.

### 3.2.3. MARS

The MARS algorithm is a non-parametric algorithm developed by [81] to estimate the complex nonlinear correlation between model inputs and output. This estimating process is achieved by automatically building a series of linear piecewise regression models through the use of basis functions, to fit the given data pair.

In the general, the MARS model is of the form precisely considered in Equation (14):

$$\hat{f}(z) = \beta_0 + \sum_{k=1}^{N} \beta_k \lambda_k(z) \tag{14}$$

where $\hat{f}(z)$ signifies the estimated output parameter value, $\beta_0$ is constant, $\lambda_k(z)$ is the *kth* basis function, $\beta_k$ signifies the *k*th basis function's coefficient and $z$ signifies the input variable. The basis function act as a hinge function to split the data into separate sections, which can be modelled individually. Each basis function can be precisely considered in Equation (15) as:

$$\lambda_k(z) = \prod_{i=1}^{I_k} \left[s_{ik} \cdot \left(z_{v(i,k)} - h_{ik}\right)\right]_+ \tag{15}$$

where $I_k$ is the quantity of splits that formed $\lambda_k(z)$, $s_{ik}$ is the selected sign with value $\pm$, $v(i, k)$ labels the predictor variable and $h_{ik}$ is the knot value on the corresponding input variables.

The MARS algorithm adopts two main steps namely: the forward selection process and the backward deletion process; to develop its model. In the forward selection process, the model is initially constructed with a constant basis function. New pairs of basis functions are thereafter iteratively included in the model to reduce the training residual sum-of-squares error; to improve the model. However, as many basis functions are added in the forward process; the model built becomes overfit and cannot generalize well with unseen data. The backward deletion process is then introduced to remove all redundant basis functions. It employs the generalized cross-validation (GCV) Equation (16) to evaluate the performance of individually created models as it eliminates the unwanted basis functions.

The individually created model with the least value of GCV is then chosen as the optimal MARS model.

$$\text{GCV}(Q) = \frac{\frac{1}{H}\sum_{j=1}^{H}\left(y_j - \hat{f}_Q(z_j)\right)^2}{\left(1 - \frac{C(Q)}{H}\right)^2} \tag{16}$$

where $y_j$ and $\hat{f}_Q(z_j)$ denotes the actual output and predicted values of the training samples, and $H$ represents the total number of training samples. As shown in Equation (17), $C(Q)$ is a penalty for model complexity that is proportional to the model's number of basis functions.

$$C(Q) = (Q+1) + pQ \tag{17}$$

where $p$ is the penalty cost for the optimization of every single basis function which works as a smoothing variable. The details of MARS as well as the selection of the $p$ are in [76].

### 3.2.4. MVRA

MVRA is a statistical tool applied to fit a model to establish a linear relation between a set of input parameters (independent variables) and an output parameter (dependent variable) [82]. This fitted model can then be used to make predictions on new data. MVRA works by studying the correlation between the various input parameters and output parameters to construct simultaneous equations so as to acquire the best-fit equation. It uses an ordinary least squares fit on the dataset to find the best-fit equation. It forms a regression matrix in the process of solving simultaneous equations. The regression matrix is then solved using the backslash operator to obtain the regression coefficient as well as the intercept [83]. Generally, the MVRA is mathematically expressed in Equation (18) as:

$$Y = \beta_0 + \beta_2 X_2 + \beta_3 X_3 + \ldots + \beta_k X_k \tag{18}$$

where $\beta_1, \ldots, \beta_k$ are the regression coefficients, $\beta_0$ is the intercept $X_1, X_2, \ldots, X_k$ is the independent variable and $Y$ is the dependent variable.

### 3.2.5. ELM

In 2004 Huang introduces the mathematical model of ELM. The ELM's basic principle is based on a single hidden layer feed-forward neural network (SLFN) (Figure 6). Because of its improved generality, simplicity, and efficient forecasting nature, the ELM has been employed in a variety of application areas [84].

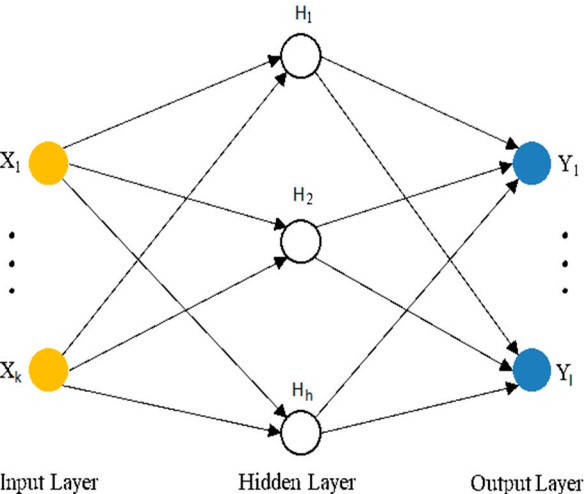

**Figure 6.** ELM Architecture.

The basic premise of ELM is as follows: Given $N$ as the number of hidden units, $K$ as the number of training samples, and the activation function $f(\ )$ in the hidden units, the output of the ELM $o_m$ for the $m$th training sample is depicted in Equation (19) as:

$$o_m = \sum_{i=1}^{N} \beta_i f(w_k, b_i, x_m) \ \ m = 1, \ldots, K \tag{19}$$

where $b_i$ is the hidden neurons' bias factor, $x_m$ denotes the number of inputs, $\beta_i$ denotes the output weight vectors and $w_k$ denotes input weight vectors. The sigmoid function is used as an activation function. The sigmoid function's output is essentially in a range of 1 to 0. To determine the output weights, the linear equation (Equation (20)) is employed.

$$\beta = H^\dagger Y \tag{20}$$

where $H$ denotes the output matrix of the hidden layer, $H^\dagger$ the Moore–Penrose generalized inverse [85] of $H$, and $Y$ denotes the ELM output targets. In Equation (21), Equation (20) is written as:

$$H\beta = Y \tag{21}$$

Equation (22) can be used to define $H$, $\beta$, $Y$ as follows:

$$H = \begin{bmatrix} p(x_1) \\ \vdots \\ p(x_K) \end{bmatrix} = \begin{bmatrix} f(w_1, b_1, x_1) & \cdots & f(w_N, b_N, x_1) \\ \vdots & \cdots & \vdots \\ f(w_1, b_1, x_j) & \cdots & f(w_N, b_N, x_j) \end{bmatrix}_{N \times K}, \beta = \begin{bmatrix} \beta_1^Y \\ \vdots \\ \beta_N^Y \end{bmatrix} \text{ and } Y = \begin{bmatrix} y_1^Y \\ \vdots \\ y_K^Y \end{bmatrix} \tag{22}$$

In this case, the hidden layer's feature mapping is $p(x)$. $H$ is the ELM's output.

### 3.3. Procedures for Model Construction

#### 3.3.1. Data Selection and Division

In modelling the various approaches presented in this study, the hold-out cross-validation technique was employed to partition the entire 101 datasets. The datasets were split into the 80:20 ratio. The first 80% of the total datasets were used as the training set (representing 81 training datasets). The remaining 20% (representing 20 datasets) were used as the test set. This strategy was adopted because [86,87] have proved that a ratio of 80:20 or 70:30 will produce accurate prediction results and will not cause overfitting.

#### 3.3.2. Data Normalization

In the data preparation phase, it is expedient that the input parameters be normalized. This is because the input parameters have different input ranges order and those with the higher values have the potential to skew the prediction results to themselves. Thus, to avoid this predicament and give equal chances to each input parameter to influence the prediction outcome, the input parameters defined in Table 1 were normalized into the interval [–1,1] [88,89] utilizing Equation (23).

$$F_i = F_{\min} + \frac{(E_i - E_{\min}) \times (F_{max} - F_{\min})}{(E_{max} - E_{\min})} \tag{23}$$

where $E_i$ signifies the actual data, $E_{max}$ and $E_{\min}$ refer to the maximum values and minimum of the actual data, $F_i$ are the normalized data and $F_{\min}$ and $F_{max}$ being the min-max values of $-1$ and 1 in that order.

#### 3.3.3. Model Development

For the development of the GPR model, five different models based on the squared exponential (Equation (24)), exponential (Equation (25)), rational quadratic (Equation (26)),

matérn 3/2 (Equation (27)), and matérn 5/2 (Equation (28)), covariance function as well as the functions were developed. Each model had a constant MF.

$$g(r, r') = v_s^2 exp\left[\frac{-\|r - r'\|}{2\chi^2}\right] \tag{24}$$

$$g(r, r') = v_s^2 exp\left[\frac{-\|r - r'\|}{\chi}\right] \tag{25}$$

$$g(r, r') = v_s^2 exp\left[\frac{-\|r - r'\|}{2\beta\chi^2}\right]^{-\beta} \tag{26}$$

$$g(r, r') = v_s^2 exp\left[1 + \frac{\sqrt{3}\|r - r'\|}{\chi}\right] exp\left[-\frac{\sqrt{3}\|r - r'\|}{\chi}\right] \tag{27}$$

$$g(r, r') = v_s^2 exp\left[1 + \frac{\sqrt{5}\|r - r'\|}{\chi} + \frac{5\|r - r'\|^2}{3\chi^2}\right] exp\left[-\frac{\sqrt{5}\|r - r'\|}{\chi}\right] \tag{28}$$

where $\beta$ is the rational quadratic covariance's shape parameter, $\chi$ is the length scale, and $v_s^2$ is the covariance function's signal variance.

The model with the lowest mean squared error and highest correlation coefficient on the test dataset was chosen as the optimum GPR model. For the BPNN model, a three-layered architecture was chosen—the first with the input layer, the second with a hidden layer and the thirdly with an output layer. A single hidden layer was used because it has been established to be a reliable predictor for any prediction problem [90]. Furthermore, in the case of hidden and output layers, hyperbolic and linear transfer functions were selected and used. The Levenberg–Marquardt algorithm was used to train this BPNN model. According to the suggested values by the previous researchers, a range of 1 to 40 for neurons was tried and the optimum number was the one that gives the lowest MSE on the test dataset [91,92]. The optimum number of neurons in the hidden layer that resulted in the lowest MSE on the test dataset was determined using a sequential experimental procedure in the construction of the ELM model. In that regard, 1 to 20 neurons were tried. It is worth stating that, the building of the MARS model, entails the choice of the highest number of basis functions to be used in the forward selection stage as well as the maximum degree of interaction. These serve as constraints in the development process. Based on their levels of interaction, three independent MARS models were built in this study–zero-degree, first degree and second-degree. Furthermore, a maximum of 20 basis functions were selected for the forward selection stage. The model with the highest correlation coefficient and lowest mean squared error (MSE) was chosen as the optimum MARS model. The MVRA model was developed using the same dataset for the development and testing of the GPR, BPNN, ELM and MARS models. The MVRA solves the multilinear regression equations established for the various input parameters and PPV using the least square technique in order to find the regression coefficient (Equation (18)) for each input parameter as well as the intercept.

### 3.3.4. Performance Indicators

The performance of the various models constructed in this study was assessed using performance measures such as variance accounted for (VAF), correlation coefficient ($R$), coefficient of determination ($R^2$) and mean squared error (MSE). These indicators are precisely shown in Equations (29)–(32) as:

$$\text{MSE} = \frac{1}{p}\left[\sum_{i=1}^{p}(s_i - q_i)^2\right] \tag{29}$$

$$R = \frac{\sum\limits_{i=1}^{p} (s_i - \bar{s})(q_i - \bar{q})}{\sqrt{\sum\limits_{i=1}^{p} (s_i - \bar{s})^2} \times \sqrt{\sum\limits_{i=1}^{p} (q_i - \bar{q})^2}} \tag{30}$$

$$R^2 = \left[ \frac{\sum\limits_{i=1}^{p} (s_i - \bar{s})(q_i - \bar{q})}{\sqrt{\sum\limits_{i=1}^{p} (s_i - \bar{s})^2} \times \sqrt{\sum\limits_{i=1}^{p} (q_i - \bar{q})^2}} \right]^2 \tag{31}$$

$$VAF = \left( 1 - \frac{var(s_i - q_i)}{var(s_i)} \right) \tag{32}$$

where $\bar{q}$ represents the mean of the estimated values, $q_i$ represents the estimated values, $s_i$ represents the measured values, $p$ is the number of observations, while $\bar{s}$ denotes the average of the measured values.

## 4. Results and Discussion

### 4.1. Developed Models

#### 4.1.1. Gaussian Process Regression

As shown in Table 6, the optimum GPR model that produced the MSE of 0.0903 and the highest *R*-value of 0.9986 for the testing dataset, had a matérn 3/2 covariance function with a noise variance of 0.06434, a length scale of 3.6019, and a signal variance of 7.0339. This indicates that the GPR-matérn 3/2 can generalize well with unseen datasets relative to the other GPR models. Hence, GPR-matérn 3/2 model was selected as the best GPR model in this study.

**Table 6.** Results of the Five different GPR Models.

| Covariance Functions | Training | | Testing | |
|---|---|---|---|---|
| | **R** | **MSE** | **R** | **MSE** |
| Matérn 3/2 | 0.9961 | 0.0798 | 0.9986 | 0.0903 |
| Matérn 5/2 | 0.9978 | 0.0452 | 0.9956 | 0.1546 |
| Squared exponential | 0.9978 | 0.0458 | 0.9942 | 0.1812 |
| Rational quadratic | 0.9978 | 0.0458 | 0.9942 | 0.1812 |
| Exponential | 1.0000 | 0.0000 | 0.9850 | 0.3008 |

#### 4.1.2. BPNN

As shown in Table 7, the optimal BPNN model has one neuron in the hidden layered network. Thus, having an architecture [7-1-1] which means seven input parameters and one neuron in the hidden layer, and an output layer. This is because it has the lowest MSE value on test datasets.

**Table 7.** Results of BPNN for Different Architectures.

| Architecture | Number of Neurons in Hidden Layer | Training | | Testing | |
|---|---|---|---|---|---|
| | | **R** | **MSE** | **R** | **MSE** |
| 7-1-1 | 1 | 0.9929 | 0.1453 | 0.9924 | 0.1714 |
| 7-2-1 | 2 | 0.9956 | 0.0902 | 0.9909 | 0.2085 |
| 7-4-1 | 4 | 0.9680 | 0.6452 | 0.8312 | 3.8234 |
| 7-5-1 | 5 | 0.9247 | 1.4831 | 0.9699 | 0.5622 |
| 7-6-1 | 6 | 0.9995 | 0.0105 | 0.4489 | 156.8569 |
| 7-7-1 | 7 | 1.0000 | 0.0007 | 0.9830 | 0.3294 |
| 7-8-1 | 8 | 1.0000 | 0.0002 | 0.9536 | 0.9092 |

**Table 7.** *Cont.*

| Architecture | Number of Neurons in Hidden Layer | Training | | Testing | |
|---|---|---|---|---|---|
| | | *R* | MSE | *R* | MSE |
| 7-10-1 | 10 | 1.0000 | $1.0244 \times 10^{-21}$ | 0.2794 | 83.2970 |
| 7-15-1 | 15 | 1.0000 | $1.97866 \times 10^{-22}$ | 0.9293 | 2.2797 |
| 7-20-1 | 20 | 1.0000 | $5.7607 \times 10^{-24}$ | 0.9008 | 2.4538 |
| 7-24-1 | 24 | 1.0000 | $1.5760 \times 10^{-23}$ | 0.8714 | 5.0448 |
| 7-28-1 | 28 | 1.0000 | $4.9485 \times 10^{-25}$ | 0.7352 | 6.7391 |
| 7-30-1 | 30 | 1.0000 | $3.6328 \times 10^{-26}$ | 0.7831 | 5.6888 |
| 7-34-1 | 34 | 1.0000 | $5.7972 \times 10^{-20}$ | 0.8136 | 8.5761 |
| 7-38-1 | 38 | 1.0000 | $9.7338 \times 10^{-26}$ | 0.5893 | 7.7243 |
| 7-40-1 | 40 | 1.0000 | $1.6407 \times 10^{-25}$ | 0.6707 | 16.7280 |

### 4.1.3. MARS

As shown in Table 8, the developed MARS model with the first order of interaction had the highest R values as well as the lowest MSE values on both the training and test datasets. Hence it was chosen as the optimum MARS model in this study.

**Table 8.** Results of Different MARS Models.

| Interaction Order | Training | | Testing | |
|---|---|---|---|---|
| | *R* | MSE | *R* | MSE |
| Zero Order | 0.9924 | 0.1548 | 0.9895 | 0.2605 |
| First Order | 0.9944 | 0.1145 | 0.9953 | 0.1038 |
| Second Order | 0.9940 | 0.1220 | 0.9923 | 0.1506 |

In the developmental process of the selected first order of interaction MARS model, only eight basis functions after the backward elimination stage were used out of the 20 basis functions employed in the forward selection stage. The eight basis functions of the selected MARS model and their respective equations are shown in Table 9.

**Table 9.** The Relationship Between Basis Functions and their Related Equations.

| Basis Function | Equation |
|---|---|
| BF1 | max (0, D – 850) |
| BF2 | max (0, 850 – D) |
| BF3 | max (0, D – 550); |
| BF5 | max (0, MC – 96.764); |
| BF6 | max (0, 96.764 – MC) |
| BF7 | max (0, D – 1750) × BF5; |
| BF10 | max (0, MC – 119) × BF3; |
| BF11 | max (0, 119– MC) × BF3; |

The developed optimum MARS model for predicting ground vibration as a result of blasting is provided in Equation (33).

$$
\begin{aligned}
PPV = {} & -2.85717 - (0.0211305 \times BF1) + (0.0270673 \times BF2) + (0.0190881 \times BF3) \\
& +(0.033926 \times BF5) - (0.0570272 \times BF6) + (5.46015 \times 10^{-5} \times BF7) \\
& +(3.56504 \times 10^{-5} \times BF10) + (2.79304 \times 10^{-5} \times BF10)
\end{aligned}
\tag{33}
$$

### 4.1.4. ELM

With respect to the experimental results shown in Table 10, the optimum ELM model developed had 12 neurons in the hidden layer with a sigmoid activation function. Thus,

having a structure [7-12-1] that represents seven inputs with 12 neurons in the hidden layer and one output.

**Table 10.** Training and Testing *R* and MSE Results for ELM.

| Architecture | Number of Hidden Neurons | Training | | Testing | |
|---|---|---|---|---|---|
| | | *R* | MSE | *R* | MSE |
| 7-1-1 | 1 | 0.6989 | 5.2395 | 0.8328 | 2.9447 |
| 7-2-1 | 2 | 0.7562 | 4.3848 | 0.8797 | 2.3808 |
| 7-5-1 | 5 | 0.9371 | 1.2477 | 0.9877 | 0.4775 |
| 7-8-1 | 8 | 0.9441 | 1.1130 | 0.9624 | 0.7056 |
| 7-10-1 | 10 | 0.9910 | 0.1832 | 0.9948 | 0.1832 |
| 7-12-1 | 12 | 0.9958 | 0.0870 | 0.9957 | 0.1384 |
| 7-15-1 | 15 | 0.9950 | 0.2181 | 0.9930 | 0.1521 |
| 7-18-1 | 18 | 0.9836 | 0.3341 | 0.9914 | 0.1989 |
| 7-20-1 | 20 | 0.9848 | 0.3080 | 0.9862 | 0.2530 |
| 7-25-1 | 25 | 0.9919 | 0.1656 | 0.9738 | 0.7639 |

4.1.5. MVRA

The developed MVRA model hsa an *R*-value of 0.7909 for the training dataset and 0.8310 for the test dataset. With respect to the MSE, the developed MVRA model had a value of 3.8341 for the training dataset and 3.2456 for the test dataset. Thus, the developed MVRA model using the training datasets for this study is shown in Equation (34).

$$PPV = 7.237178 + 0.714419AD - 2.80436B + 3.443905S - 0.02705MC$$
$$-2.33861SL - 0.67419PF - 0.00284D \tag{34}$$

*4.2. Assessment of Models Performance*

In evaluating the prediction capabilities of the five predictive models presented in the study, the statistical performance outcomes of the testing samples are outlined in Table 11.

**Table 11.** PPV Prediction Results of Various Models.

| Model | MSE | *R* | $R^2$ | VAF (%) |
|---|---|---|---|---|
| GPR | 0.0903 | 0.9985 | 0.9971 | 99.1728 |
| MARS | 0.1038 | 0.9953 | 0.9906 | 98.8692 |
| ELM | 0.1381 | 0.9957 | 0.9915 | 98.5469 |
| BPNN | 0.1714 | 0.9924 | 0.9848 | 98.2273 |
| MVRA | 3.2456 | 0.8310 | 0.6906 | 66.0603 |

Notionally, a predictive model is said to be accurate if *R* and $R^2$ are 1, MSE is 0 and VAF is 100%. In that regard, it can the seen that the GPR with the MSE value of 0.0903 closest to 0, *R* values of 0.9985 closest to 1, $R^2$ values of 0.9971 closest to 1 and VAF value of 99.1728% closest to 100% outperformed all the techniques applied in this study. This shows the reliability of the GPR in predicting ground vibration. The MARS performed better than the ELM by having had MSE value of 0.1038 and a VAF value of 98.5469% with the ELM having an MSE value of 0.1381 and a VAF value of 98.2273%. The ELM also performed better than the BPNN with MSE and VAF values of 0.2178 and 98.1919%. It is worth mentioning that the GPR, MARS, ELM and BPNN were superior in predicting ground vibration to the simple MVRA model which had an MSE of 3.2456, *R*-value of 0.8310, the $R^2$ value of 0.6906 and VAF value of 66.0603%. Figure 7 depicts the interpretation of the obtained results.

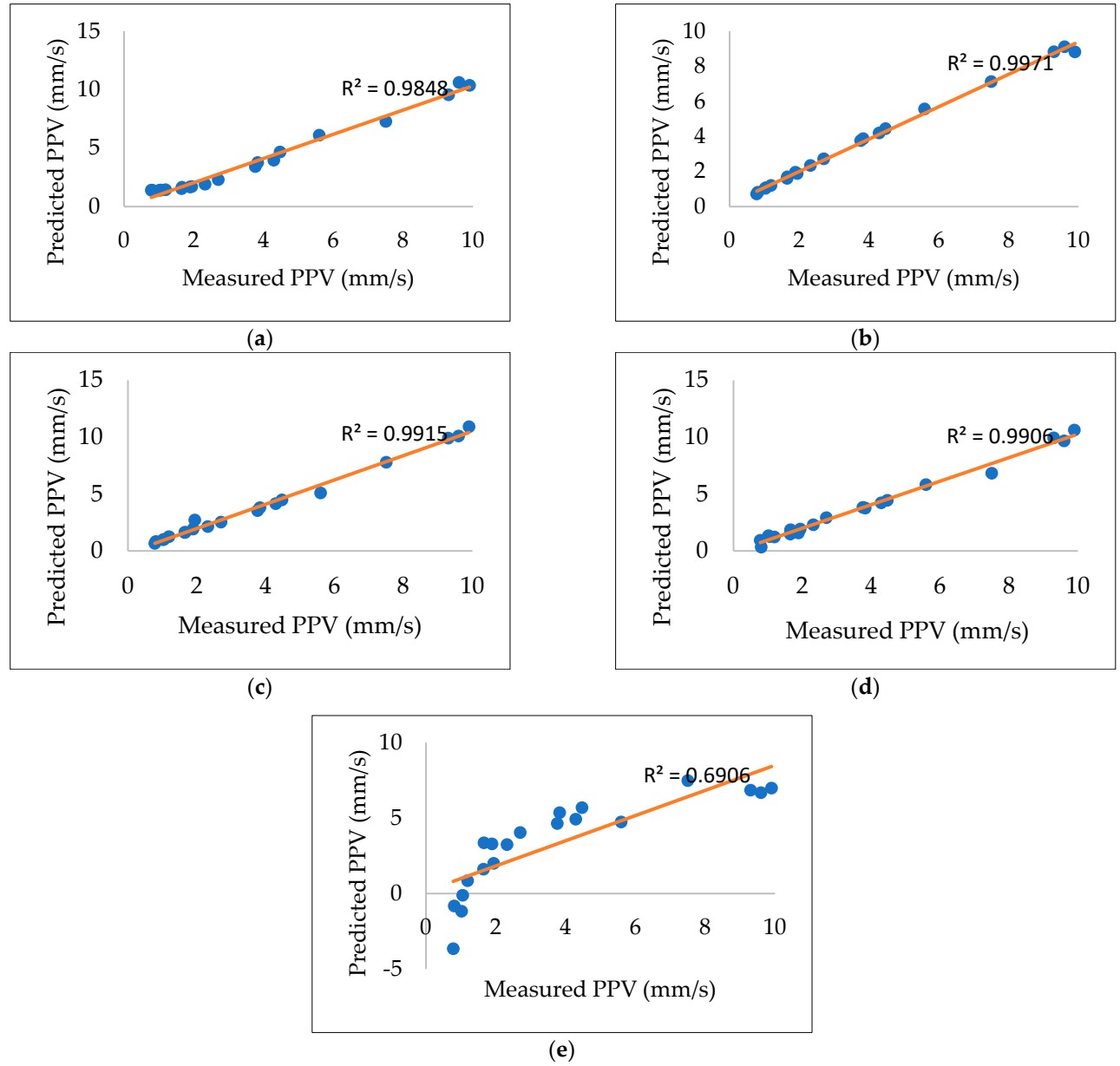

**Figure 7.** Comparison of Predicted and Measured PPV for: (**a**) BPNN (**b**) GPR (**c**) ELM (**d**) MARS (**e**) MVRA.

As ground vibration is one of the most unfavorable environmental effects of blasting operations which can cause serious damage to neighboring residences and structures, a precise prediction of its severity is critical to managing and lessening its incidence. The $R$, $R^2$ and VAF values for the GPR, MARS, BPNN, and ELM may not vary significantly, but any predictive model that delivers the most accurate prediction is of paramount relevance to the blast engineer. Hence the need to develop different models. This study found that the GPR is more accurate in forecasting ground vibration than the MARS, BPNN, ELM and MVRA and that it can be used by blast engineers to predict blast-induced ground vibration.

### 4.3. Sensitivity Analysis

To determine the most and least effective parameters, sensitivity analysis is performed to examine how the model responds to changes in the input variables with respect to PPV. Hence, in this study, a sensitivity analysis approach implemented in [93] was adopted. Here, while keeping the ranges of all other parameters fixed, the mean value of one of the

input variables is increased (i.e., New mean = Old mean + 5% Old Mean) and subsequently the amount of changes in the predicted PPV using the GPR model is recorded. The obtained results are graphically illustrated in Figure 8.

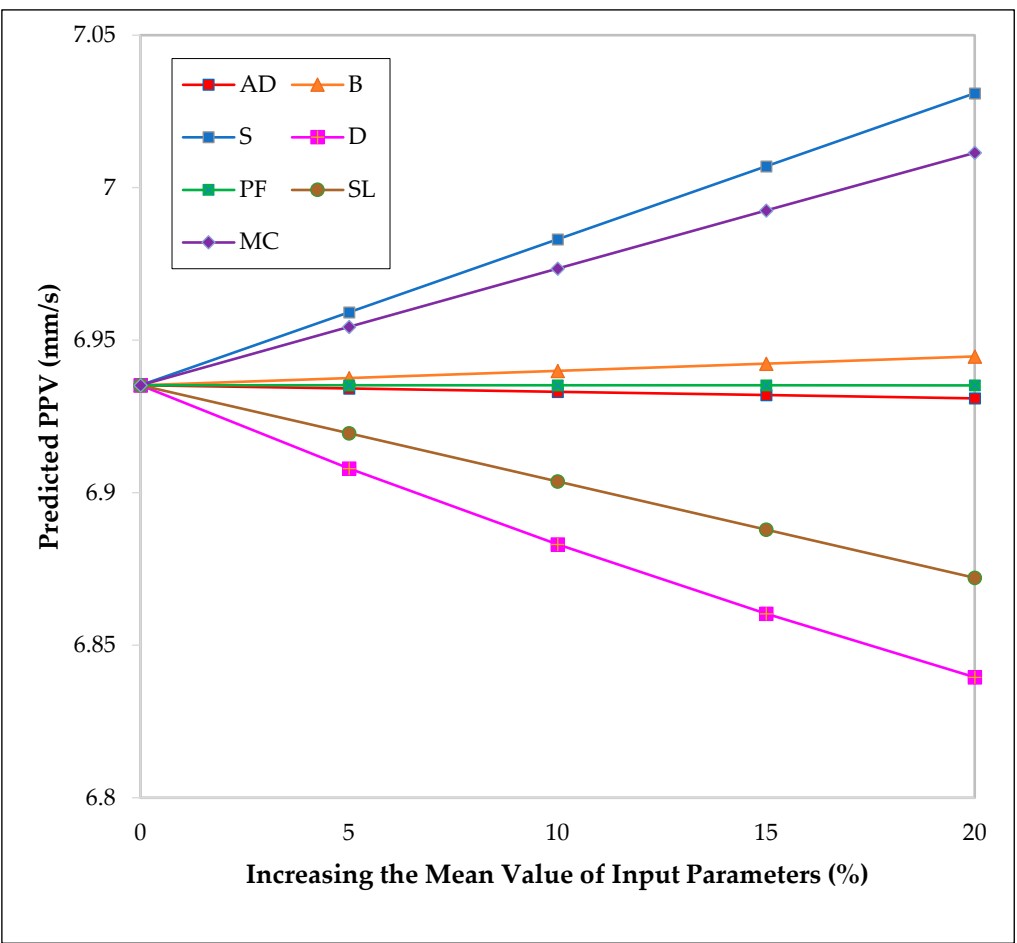

**Figure 8.** The Input Parameters and PPV Relationship's Strength.

As can be seen in Figure 8, increasing the mean values of spacing and maximum charge per delay, increases PPV. Furthermore, increasing the mean values of distance and stemming length decreases PPV. Increasing burden slightly increased PPV. Nevertheless, increasing values of powder factor and average hole depth did not significantly impact the values of PPV. It can thus be said that the most influential parameters that can affect PPV greatly are spacing, a maximum charge per delay, distance and stemming length.

## 5. Conclusions

In this paper, three AI models of GPR, ELM and BPNN were developed and applied to predict blast-induced PPV. In showing the predictive capabilities of these AI techniques, a MARS and MVRA model were also developed. In developing and evaluating these models, 101 datasets obtained from Ras Limestone Mine of Shree Cement, India were utilized. Out of the 101 datasets, 81 were utilized to create the various models, while the remaining 20 were used as test sets for the models that were developed. The input parameters in the creation of the various models were average depth (m), burden (m), spacing (m), powder factor (t/kg), the distance between the monitoring station and the blasting site (m), stemming length (m), and maximum charge per delay (kg), while the output parameter was PPV. The various developed models were then evaluated using performance metrics of $R$, $R^2$, MSE and VAF. The results obtained showed that the GPR model had the lowest MSE of 0.0903, and the highest $R$, $R^2$, and VAF values of 0.9985, 0.9971 and 99.1728% respectively,

indicating that it was superior to the other models in predicting blasting-induced ground vibration. This was followed by MARS which had MSE, $R$, $R^2$ and VAF values of 0.1038, 0.9953, 0.9906 and 98.8692% respectively. Then ELM had an MSE of 0.1381, $R$-value of 0.9957, $R^2$ value of 0.9915 and VAF value of 98.5469. Then the BPNN with an MSE, $R$, $R^2$ and VAF of 0.1714, 0.9924, 0.9848 and 98.2273% respectively. The MVRA performed very poorly as it had, with the highest MSE of 3.2456, and lowest $R$-value of 0.8310, the $R^2$ value of 0.6906 and the VAF value of 66.0603%. The results obtained show that the GPR model can be utilized to forecast blast-induced ground vibration in the mining industry. The sensitivity analysis of the dataset found that spacing, a maximum charge per delay, distance and stemming length had a great influence on PPV whereas burden, powder factor and average depth had slight to no influence on PPV.

**Author Contributions:** Conceptualization, E.T.M., R.M.B., C.K.A., M.K.; methodology, R.M.B., C.K.A.; software, R.M.B., C.K.A.; formal analysis, R.M.B., C.K.A.; resources, E.T.M., R.M.B., C.K.A.; data curation, R.M.B. writing—original draft, M.B., E.T.M., R.M.B., C.K.A., M.K., M.M.S.S., S.K.; writing—review and editing, M.B., E.T.M., R.M.B., C.K.A., M.K., M.M.S.S., S.K.; Supervision, E.T.M., M.K., S.K.; funding acquisition, M.M.S.S. All authors have read and agreed to the published version of the manuscript.

**Funding:** The research is partially funded by the Ministry of Science and Higher Education of the Russian Federation under the strategic academic leadership program 'Priority 2030' (Agreement 075-15-2021-1333 dated 30 September 2021).

**Institutional Review Board Statement:** Not applicable.

**Informed Consent Statement:** Not applicable.

**Data Availability Statement:** The data presented in this study are available on request from the corresponding author.

**Acknowledgments:** Authors are thankful to Pankaj Agarwal, Assistant Vice President and Management of Shree Cement, Beawar, Rajashthan for providing data for the preparation of this paper.

**Conflicts of Interest:** The authors declare no conflict of interest.

## Abbreviations

The following abbreviations are used in this manuscript:

| Abbreviations | Explanations |
| --- | --- |
| ABC | Artificial bee colony |
| ANN | Artificial neural network |
| BA | Bat-inspired Algorithm |
| BN | Bayesian network |
| BBO | Biogeography-based optimization |
| BI | Blastability index (compressive strength/tensile strength) |
| BIENN | Brain-inspired emotional neural network |
| B | Burden |
| BS | Burden spacing ratio |
| CHAID | Chi-square automatic interaction detector |
| CART | Classification and regression tree |
| H | Distance between blasting face and monitoring point (m) |
| XGBoost | Extreme gradient boosting machine |
| ELM | Extreme learning machine |
| FFA | Firefly algorithm |
| FIS | Fuzzy inference system |
| FL | Fuzzy logic |
| GPR | Gaussian process regression |
| GEP | Gene expression programming |
| GA | Genetic algorithm |
| GP | Genetic programming |

| GOA | Grasshopper optimization algorithms |
|------|-------------------------------------|
| GWO | Grey wolf optimization |
| GMDH | Group method of data handling |
| HHOA | Harris hawk optimization algorithm |
| HD | Hole depth |
| ICA | Imperialistic competitive algorithm |
| KNN | K-nearest neighbors |
| LSSVM | Least square support vector machine |
| M5DT | M5′ decision tree |
| Q | Maximum charge per delay |
| MARS | Multivariate adaptive regression splines |
| ANFIS | Neuro-fuzzy inference system |
| NH | Number of holes |
| N | Number of rows |
| ORELM | Outlier robust ELM |
| PSO | Particle swarm optimization |
| V | Poisson's ratio |
| PF | Powder factor |
| $P_v$ | P-wave velocity |
| RF | Random forest |
| RVR | Relevance vector regression |
| RSM | Response surface methodology |
| RD | Rock density |
| RQD | Rock quality designation |
| SaDE | Self-adaptive differential evolution |
| S | Spacing (m) |
| ST | Stemming length |
| SD | Subdrilling |
| SVM | Support vector machines |
| SVR | Support vector regression |
| VOD | Velocity of detonation |
| WNN | Wavelet neural network |
| WOA | Whale optimization algorithm |
| E | Young's modulus |

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
