# Peer review of "Prediction of Blast-Induced Ground Vibration at a Limestone Quarry: An Artificial Intelligence Approach"

_applsci, doi:10.3390/app12189189_

Round 1

Reviewer 1 Report

The manuscript applsci-1862499 describes an analysis of signals from blast-induced ground vibrations in a limestone quarry. The authors use three statistical methods, referred to as "Artificial Intelligence Approaches", to investigate whether the main parameter that characterizes ground vibrations, namely the peak particle velocity, can be predicted. They compare the predictions of these AI approaches to those of two methods based on regressions.

I recommend the publication of this manuscript. However, the authors should address several editorial and technical issues.

EDITORIAL

The paper is very long. There are many instances of repetitions, double definitions and unnecessary sentences or details. I feel that a re-editing that would suppress these defects could reduce the manuscript length by 20-%. It is not possible to list them all, but here are some:

Abstract

P1/L23: "Though" should be suppressed.
P1/L29: "which is an AI technique" is unnecessary because GPR is defined at L30 above.

1. Introduction

- The number of acronyms is huge. Are there all necessary? Typically, an acronym for a word group is useful over four repetitions of the group throughout the paper. If not, group repetition is OK. Consider a table for the necessary acronyms.

- P2/L69: separate "the" and "1950s".

- P2/L84 and 85: The author names should begin the sentences, not the reference numbers.

- P2/L79 and P3/L99: double definition of "particle swarm optimization (PSO)".

- P3/L104 and 113: double definition of "extreme gradient boosting machine (XGBoost) ".

- There are several uses of "holistic" and "holistically" which, to me, are out of their usual field of usage in this article, if not misused.

- Here is an example of re-editing that reduces the word count using the last three sentences of the introduction: "In that regard, this study is exploratory. The models described above consider seven effective parameters, namely the average depth, maximum charge per delay, powder factor, spacing, burden, distance and stemming length, because, as shown in [5–7], they significantly affect the intensity of ground vibration."

3. Methodology

- The in-line maths print above the baseline throughout the document.

- There are many math symbols. Consider a table.

TECHNICAL

- Unless I misunderstand, all data sets were obtained from the same quarry, so they may show small parametric variations. Can the authors comment on the sensitivity of the considered AI approaches to larger variations of their parameters?

- Several wave techniques now offer efficient tools to obtain 3D- mapping of grounds. Computer codes such as DYNA include constitutive relations that describe well rock fracturing and ground deformations, and propose a large library of non-ideal explosives. Can the authors comment on how the AI approaches would compare with numerical simulations using ground mapping as initial data and on the effectiveness conditions of such simulations?

Reviewer 2 Report

The manuscript has 24 pages, 84 citation references, 8 figures. The manuscript is about the artificial methods (AI) in Ground vibration prediction as effects of quarry blasting activities as an example of Shree Cement Ras Limestone Mine in India. The authors tried to show all AI methoвы and correlation for Peak Particle Velocity (PPV), using input parameters ST stemming length (m), S spacing (m), B burden (m), HD hole depth (m), BS burden spacing ratio, SD subdrilling (m). The results of Comparison of Predicted and Measured PPV for: (a) BPNN (b) GPR (c) ELM (d) MARS (e) MVRA could help for the prediction model choice. 

Comments

Authors’ affiliations: 2 and 3 are the same, so no need to enumerate 3 separately. It is 2. The 5 and 7 – index of the area (ZIP-code).

Line 32 Shree Cement Ras Limestone Mine in India – it is a full proper name.

There are too many repetitions like lines 88 and 89: “… [32] to predict flyrock. [33] applied …to predict flyrock. [34] applied … to predict flyrock”. Lines 97 and 99 “for predicting ground vibration”. Lines 101& 102&109&111&113&116&117“for estimating ground vibration”. L. 114&115 “built a hybrid model”. This is categorically unreadable. To avoid repetition, the authors need to make a comparative table and sum-up all tools.

Check all methods name (AOp or AOP) (k-nearest or K-nearest). Line 325 p(x) in italic

Table 1. comma between MARS ANN

Figure 1 and 2 have the same titles. Figure 1 should be called Figure 1. Blasting Round View shema and Figure 2. Blasting Round View Shree Cement Ras Limestone Mine in India.

L. 168 put the table 2 here for the first mention.

L.174 an Instantel Micromate ISEE Std/XM seismograph reference?

Table 2. Description of dataset parameters – are they input parameters? Or not? So why Peak Particle Velocity (PPV ) is here as an input parameter?

Fig. 3 bad quality

Fig. 4 title my suggestion “Figure 4. Portable ground vibration monitoring station in realistic conditions (at Shree Cement Ras Limestone Mine in India)”. The figure title needs to be near the figure, not at the next page

L. 189-192 jumps into Peak Particle Velocity (PPV ) and other input parameters. It is the inexplicable appearance of the PPV importance. Then I see the formulae (33) in L. 422 – the first time I understand, that for vibration prediction authors decide to look for PPV. Because Ground vibration is measured in terms of Peak Particle Velocity (PPV) with units in mm/s or mm/s. I am sorry, I don't know it, as I am specialist from a different subject.

In my opinion, the Lines 220-221 Prediction using Gaussian Process (GP) should be before 3.1. Mathematical Description of the Different Methods. Because you explain, how you can predict.

L. 314 hidden units f( ), what are these hidden units (layers)?

The subsection 3.2 should be moved up before explanation of methods.

Figure 6. A Systematic Flowchart for Prediction of Blast-Induced Ground Vibration

L. 351 [81] why is so old citation? and only one?

S. 4.1.1 full name here, please,  Gaussian process regression

Table 9. Results of Various Models. Only for PPV?

Figure 7 has R values with unnecessary excessive number of characters after the comma

Final: if the research is supported by the Russian program 'Priority 2030', why there are no references to the Russian sources? It seems to me, that it out of the frame idea. There should be some. Examples: Grishchenkova, E.N. Development of a Neural Network for Earth Surface Deformation Prediction. Geotech Geol Eng 36, 1953–1957 (2018). https://doi.org/10.1007/s10706-017-0438-y ; Isheyskiy V., Marinin M., Dolzhikov V. Combination of fracturing areas after blasting column charges during destruction of rocks. International Journal of Engineering Research and Technology. 2019. Vol. 12(12), p. 2953-2956; Kholodilov A.N., Gospodarikov A.P. Modeling Seismic Vibrations under Massive Blasting in Underground Mines. Journal of Mining Science. 2020. Vol. 56. Iss. 1, p. 29-35. DOI: 10.1134/S1062739120016454; [MDPI series] Koteleva, N.; Frenkel, I. Digital Processing of Seismic Data from Open-Pit Mining Blasts. Appl. Sci. 202111, 383. https://doi.org/10.3390/app11010383 Uzbekistan: Khudoyberdiev F.T., Nurboboev Y.T., Maksudov S.F., Shomurodov S.M. The process of destruction of rock by an explosion with the use of blasthole stemming in roadheading mining operation. IOP Conference Series: Earth and Environmental Science, 14-16 October, 2020, Tashkent, Uzbekistan. IOP, 2020. Vol. 614, p. 012067. DOI: 10.1088/1755-1315/614/1/012067. Tokmantsev M.S. Assessment of seismic impact on the marginal mass of the Jubilee quarry after preliminary slitting. MIAB. Mining Inf. Anal. Bull. 2015. no S4—2. p. 181—187. [In Russ]

I also think, that the citation of one of the authors is helping for the introduction section (concerning names of mathods) Bhatawdekar, R.M., Armaghani, D.J., Azizi, A. (2021). Review of Empirical and Intelligent Techniques for Evaluating Rock Fragmentation Induced by Blasting. In: Environmental Issues of Blasting. SpringerBriefs in Applied Sciences and Technology. Springer, Singapore. https://doi.org/10.1007/978-981-16-8237-7_2

I also suggest adding Applied Sciences (mdpi) publications in the reference list:

Leskovar, K.; Težak, D.; Mesec, J.; Biondić, R. Influence of Meteorological Parameters on Explosive Charge and Stemming Length Predictions in Clay Soil during Blasting Using Artificial Neural Networks. Appl. Sci. 2021, 11, 7317. https://doi.org/10.3390/app11167317

Choi, Y.-H.; Lee, S.S. Predictive Modelling for Blasting-Induced Vibrations from Open-Pit Excavations. Appl. Sci. 2021, 11, 7487. https://doi.org/10.3390/app11167487

There should be a list of all of abbreviations in the end of the manuscript.

Reviewer 3 Report

This study aims to develop predicting models for blast-induced vibrations of at a limestone quarry with application of different methods of AI. To this aim, 101 blasting datasets were used totally and 80% of this data was used for training part and the remaining was used for the validating part. The results showed that GPR method has the highest accuracy and efficiency in predicting the PPV value. The paper is well prepared and written while the following comments should be addressed in the revised manuscript before publishing the paper.

1)      There are several misspelling problems in the manuscript. As an example, the word “basting” on line 422 should be corrected and replaced with “blasting”. Also, please use a space in between “the1950s” on line 69 and “3.8341for” on line. The authors need to read again the manuscript carefully and resolve similar problems.

2)      80% of dataset was used for training and 20% for validating. Based on what strategy and reference these values were adopted by the authors? How the results will be affected by changing the values from 80 and 20 to 70 and 30%? Please comment and provide related reference.

3)      Are training and validating dataset randomly selected? If yes, what effect does random selection of values have on the results? How do the authors ensure that the proposed models has the best accuracy, efficiency and maximum level of R2? I mean that, by changing the values selected for training and validating datasets, the results also change. Please comment.

4)      The general form of PPV formulation (proposed by many researchers) is in the form of PPV=k*D^(-b) where k and b are fitting parameters and D is scaled distance of explosion. Why the author did not use this form in providing a new formulation?

5)      Several research studies can be found in the literature that shown that the GPR method is a remarkably powerful class of non-parametric machine learning algorithm and is more powerful respect to other AI algorithms. Accordingly, why the authors used five different methods in this study? Wouldn't it be better to use only the GPR method? Please comment.

6)      How can a reader predict PPV value using the results of sections 4.1.1, 4.1.2 and 4.1.4? I mean that, no one can use these results in her/his own research study, unless he/she has the ability to perform the methods used in these sections. Can the authors provide a model (similar to the relationships proposed in sections 4.1.3 and 4.1.5) that could be used to predict PPV with simplicity and easy to follow?

7)      Please check figure 7. Some data are missing on the sub-figures (on the vertical axis). Please resolve the problem in the revised manuscript.

8)      The sensitivity analysis is used to find the sensitive parameter. However, this section needed to be extended and further explanations should be given on interpreting the results of figure 8 in the revised manuscript.

9)      Can the author compare the results of their proposed method with those models proposed by other researchers in the literature? The following paper may give an idea to the authors to address this comment. New figure can be added in the revised file.

“Determination of blast-induced ground vibration equations for rocks using mechanical and geological properties”

Round 2

Reviewer 3 Report

Almost all my comments have been answered , and some of them have been addressed appropriately in the revised manuscript. However, the provided response for some comments are not convinced me. They are as follow:

Comment number 4: The reason for not using this kind of relationship should be mentioned in the revised file. Also, the authors said that there are several research studies that proved this model does not produce accurate results. Relevant references should be prepared and included in the second revised file.

Comment number 8: The part related to sensitivity analysis is not properly extended. I recommend the authors to prepare a new figure in the second revised file to show how each parameter affects the output. For example, the new figure will show by increasing one of the parameters, the PPV increases, decreases, or remains constant. Obtaining this diagram helps any readers to extract the most sensitive and non-sensitive parameter with better accuracy based on the trend of the diagram. So, from the reviewer point of view, the paper needs further improvement on section 4.3. (i.e. sensitivity analysis part). The following paper with DOI: https://doi.org/10.1016/j.engstruct.2020.110909 can be used and referred by the authors for addressing this comment in the second revised file.
